# Automatic Assessments of Parkinsonian Gait with Wearable Sensors for Human Assistive Systems

**DOI:** 10.3390/s23042104

**Published:** 2023-02-13

**Authors:** Yi Han, Xiangzhi Liu, Ning Zhang, Xiufeng Zhang, Bin Zhang, Shuoyu Wang, Tao Liu, Jingang Yi

**Affiliations:** 1The State Key Laboratory of Fluid Power and Mechatronic Systems, School of Mechanical Engineering, Zhejiang University, Hangzhou 310027, China; 2Department of Intelligent Mechanical Systems Engineering, Kochi University of Technology, Kochi 782-8502, Japan; 3The National Research Center for Rehabilitation Technical Aids, Beijing 102676, China; 4The College of Mechanical and Electrical Engineering, China Jiliang University, Hangzhou 310018, China; 5Department of Mechanical and Aerospace Engineering, Rutgers University, Piscataway, NJ 08854, USA

**Keywords:** wearable sensors, unified Parkinson’s disease rating scale, gait analysis, human assistive system

## Abstract

The rehabilitation evaluation of Parkinson’s disease has always been the research focus of human assistive systems. It is a research hotspot to objectively and accurately evaluate the gait condition of Parkinson’s disease patients, thereby adjusting the actuators of the human–machine system and making rehabilitation robots better adapt to the recovery process of patients. The rehabilitation evaluation of Parkinson’s disease has always been the research focus of rehabilitation robots. It is a research hotspot to be able to objectively and accurately evaluate the recovery of Parkinson’s disease patients, thereby adjusting the driving module of the human–machine collaboration system in real time, so that rehabilitation robots can better adapt to the recovery process of Parkinson’s disease. The gait task in the Unified Parkinson’s Disease Rating Scale (UPDRS) is a widely accepted standard for assessing the gait impairments of patients with Parkinson’s disease (PD). However, the assessments conducted by neurologists are always subjective and inaccurate, and the results are determined by the neurologists’ observation and clinical experience. Thus, in this study, we proposed a novel machine learning-based method of automatically assessing the gait task in UPDRS with wearable sensors as a more convenient and objective alternative means for PD gait assessment. In the design, twelve gait features, including three spatial–temporal features and nine kinematic features, were extracted and calculated from two shank-mounted IMUs. A novel nonlinear model is developed for calculating the score of gait task from the gait features. Twenty-five PD patients and twenty-eight healthy subjects were recruited for validating the proposed method. For comparison purpose, three traditional models, which have been used in previous studies, were also tested by the same dataset. In terms of percentages of participants, 84.9%, 73.6%, 73.6%, and 66.0% of the participants were accurately assigned into the true level with the proposed nonlinear model, the support vector machine model, the naive Bayes model, and the linear regression model, respectively, which indicates that the proposed method has a good performance on calculating the score of the UPDRS gait task and conformance with the rating done by neurologists.

## 1. Introduction

Walking training is the main rehabilitation modality for PD. An accurate PD gait assessment result can help the assistive system automatically adjust the actuators’ mode, so as to better adapt to the training needs of PD in different rehabilitation periods [1,2]. For assessing the extent of gait impairments in PD patients, the Unified Parkinson’s Disease Rating Scale (UPDRS) is a widely accepted standard in clinics [3]. The gait task in UPDRS is rated with five levels (0, 1, 2, 3, and 4) to reflect the severity, which always shows normal walking, slow walking with small steps, severe gait disorder, needs external assistance, and loss of walking ability, respectively. The assessment is commonly done by neurologists according to the corresponding descriptions about the levels. However, the descriptions are subjective and the defined boundaries are ambiguous. Because of this reason, the results of assessments can have differences between different raters [4]. For example, in [5], about 25% of PD patients were rated as having different gait task levels by two different neurologists.

As the development of sensor technology, motion capture systems and force platforms have been used as more objective and accurate approaches for evaluating PD gait than the assessments from neurologists [6,7,8]. However, because of their high cost and restricted site, the inertial measurement unit (IMU) is gradually used as an alternative in this application [9,10,11,12]. Many gait parameters, which reflect the spatial, temporal, and kinematic gait characteristics, are extracted from the measurements of IMUs attached to trunk and lower limbs for quantifying and evaluating the gait impairments in PD patients [13,14,15,16,17]. The evaluation results and the UPDRS gait task levels are found to have good consistency by some researchers [13].

To expand the applicable area in the clinic, the results of the sensor systems need to be matched with the clinical standards, such as UPDRS. For this purpose, several researchers have used the results of the sensor systems to evaluate some tasks [18,19,20,21] in the UPDRS. Various machine-learning-based classifiers such as support vector machine (SVM), decision tree (DT), and k-Nearest Neighbors (k-NN) are implemented in these studies [20]. For example, Parisi et al. [5] used three IMUs on the trunk and thighs to calculate spatial and temporal gait parameter and explored three classifiers to estimate the gait task level; they reached a similar accuracy with the judgement of a different neurologist. Patel et al. [18] used eight IMUs attached to limbs and trunk to extract features in time domain and frequency domain when patient performing specified movements and used SVM to estimate the levels of tremor task and bradykinesia and dyskinesia tasks. Giuberti et al. [19,21] used three IMUs on trunk and thighs to monitor the kinematic features during patients taking sit-to-stand and leg ability task and explored the performances of three classifiers on estimating the corresponding levels. Jeon et al. [20] used an IMU attached on finger and explored four classifiers to assess the tremor task.

However, there is still little research on the automatic scoring for gait task of UPDRS [5,22]. Federico et al. [5] developed a Body Sensor Network (BSN) consisting of 3 IMUs and discussed the effect of different classification methods for automatic scoring the gait task of UPDDRS and finally realized 53% with no error and 98% with the error not more than 1. Heldman et al. [22] implemented multiple linear regression models for quantifying the detailed scores of toe-tapping task, leg agility task, gait task, and freezing of gait task based on two heel-attached IMUs, where the average correlation coefficient and RMS error were 0.86 and 0.47, respectively. There are still some gaps in the current relevant research and actual clinical application. First, the accuracy of the current research needs to be improved, and it cannot meet the actual clinical needs. In addition, the assessments are still too inaccurate in some situations. The range of each level is defined widely in the UPDRS. The classifiers can distinguish the different levels, but they are hard to further quantify the severity in the same level. This causes the difficulty of comparing the patients who are in the same level and in monitoring the changes of a patient during a short term.

In this paper, we propose a new nonlinear model to automatically score the gait of Parkinson’s patients, corresponding to UPDRS. By recruiting 28 healthy subjects (9 healthy subjects over 50 years old and 19 healthy subjects under 50 years old) and 25 Parkinson’s patients, the scoring results were verified. The first innovation of this paper is that the scores output by the proposed model are continuous, which expands the application range of UPDRS, because even if subjects are scored into the same level, there are also varying degrees of severity. The second innovation is that compared with the results of three other commonly used classifiers (73.6%, 73.6%, and 66.0% for the support vector machine model, the naive Bayes model, and the linear regression model, respectively), the accuracy of the new nonlinear model reaches 84.9%, which improves the possibility of clinical application. The third is that the hardware system is only composed of two MPU6050 chips, which is low in cost, easy to use, and easier to integrate into the assistive system.

## 2. Methods

### 2.1. Participants and Protocols

The information of subjects is shown in Table 1, in which the healthy subjects are used to define the healthy range and regarded as level 0. Because patients in level 3 and level 4 cannot walk independently, to avoid accidents, they are not involved in our study. To better suppress the influence of irrelevant factors on the experimental results, subjects with PD were recruited from the rehabilitation department of Zhejiang Hospital for gait data collection experiments at the same time in one day from March 2021 to December 2021, all with acupuncture as treatment method. In order to control variables, all patients were graded by the same neurologist with more than 20 years of experience through UPDRS. All patient data were measured prior to treatment, as treatment can have a modifying effect on patient gait. All subjects gave their informed consent for inclusion before they participated in the study.

Each PD participant and healthy participant took a walking trial with a wearable IMU system for three times. During the trial, participant was asked to walk in a straight line at a preferred speed for more than 12 m on a flat floor. The initial four steps were regarded as the accelerating phases of walking. The following ten steps were regarded as the steady walking phase. The measurements of the system during the steady walking phase were further used for the assessments of the gait task in UPDRS. This study was approved by the Medical Ethics Committee of School of Biomedical Engineering and Instrument Science, Zhejiang University (Project identification code:2021-39).

### 2.2. The Configuration of System

As shown in Figure 1, the motion information of the subjects is collected through Wearable Sensors. Based on Feature Calculation, the gait features such as stride length from the original motion data could be obtained, which is imported into the Nonlinear Model to automated scoring of Parkinson’s patients.

### 2.3. Wearable System

The wearable system contains two Fuzhi${}^{\mathrm{TM}}$-gait (based on InvenSense MPU-6050 chip, including a 3D accelerometer with a range of ±8 g and a 3D gyroscope with a range of ±1000${}^{\circ }$/s, with a size of 25 × 40 × 15 mm, a weight of 100 g, and a sample rate of 100 Hz, which will not become a burden for the subjects to walk and was also used in [23]). Each IMU is attached close to the ankle on the lateral side of each shank.

The X-axis, Y-axis, and Z-axis are along to inferior–superior, anterior–posterior, and mediolateral direction, respectively. For the convenience on describing the outputs, a global coordinate system is introduced. The P-axis in the global coordinate system is along the direction of progression (the connection between two adjacent landing points of the limb), the V-axis is along the direction of gravity, and the L-axis is in the mediolateral direction. Each IMU’s data is sampled at a frequency of 100 Hz. For research purposes, the data of IMUs is saved into a micro SD card and processed offline on MATLAB (R2021a).

### 2.4. Feature Calculation

The outputs of the wearable IMU system are twelve gait parameters listed in Table 2, which represent the kinematic characteristics of the shank. The gait parameters are further used as inputs in the proposed model for automatically assessing the gait task in UPDRS. Most of the gait parameters have already been applied in [24,25,26] and showed good performance on gait evaluation and classification.

An algorithm developed in a previous study [24] is implemented for calculating the gait parameters of both sides by the attached IMUs. The data collected by the IMUs are first processed by a coordinate system correction algorithm [26], which is used to reduce the estimation error caused by the unaligned sensors’ coordinate system. Heel-strike (HS) and toe-off (TO) gait events are detected according to the cyclic pattern of measured z-axis angular velocity based on a method developed in [17]. HSs are used to separate the walking into gait cycles. TO is used to separate a gait cycle into stance and swing gait phases. The shank orientations and ankle motion trajectories during each gait cycle are calculated by the integration of measured angular velocity and acceleration. To reduce integrated error, zero velocity state of the ankle motion is detected in stance phase for resetting the motion calculation, and compensations are calculated for correcting the integrated error [24]. Based on the detected gait phases and ankle motion, three spatial-temporal gait parameters including stride length (SL), gait cycle duration (GD) and percentage swing phase (PSP), and nine kinematic gait parameters including max ankle height (MH), range of lateral displacement (RL), range of shank Z-axis rotation (RSZ), range of shank y-axis rotation (RSY), range of shank X-axis rotation (RSX), max ankle progressive velocity (MPV), max ankle vertical velocity (MVV), max shank Z-axis angular velocity (MSV), and ankle displacement at MH (MHD) are extracted.

Considering that some gait parameters are related with subject’s biomechanical parameters, the gait parameters SL, MH, RL, MPV, MVV, and MHD are normalized by subject’s height.

### 2.5. Nonlinear Model

#### 2.5.1. Structure

A novel multiple nonlinear model is proposed to simulate the nonlinearity in the UPDRS assessment process of neurologists and is used for automatically estimating the gait task scores from the gait features. Unlike the levels rated by neurologists, the scores output by the proposed model are continuous. Thus, the actual range of the score is set as 0 ± 0.5, 1 ± 0.5, 2 ± 0.5, 3 ± 0.5, and 4 ± 0.5 for the five levels of gait task in UPDRS, respectively. The levels of gait task have a positive correlation with the severity of gait impairment. However, they are not defined quantitatively and spaced evenly. Thus, four adjustable parallel hyperplanes are developed in the proposed model to divide the five levels. These hyperplanes are defined as
(1)$$W^{T}x+C=D_{i},i=1,2,3,4$$
where *x* is the set of the gait features, *W* and *C* are coefficients, and $D_{i}$ is a set of coefficients for differentiating the four hyperplanes. Because these hyperplanes are parallel with each other, a unified expression can be used to describe the relative position of a data point with these hyperplanes
(2)$$x_{hp}=\frac{W^{T}x+C}{\left|\right|W\left|\right|}=w^{T}x+c$$
where $x_{hp}$ is the distance of a data point to the plane $W^{T}x+C$. According to the value of $x_{hp}$, the data point can be assigned into the five UPDRS levels. The detailed score can be further calculated by the relative position of the data point with the adjacent hyperplanes.

Because the scores need to be limited by lower and upper bounds, a sigmoid function is implemented to redefine the $x_{hp}$ [27]. The redefined distance (yrd) is calculated as
(3)$$y_{rd}=\frac{b_{u}-b_{l}}{1+e^{-(x_{hp}-\frac{b_{u}+b_{l}}{2})}}+b_{l}=\frac{5}{1+e^{-x_{hp}+2}}-0.5$$
where $b_{l}$ and $b_{u}$ are the lower and upper bounds of the output score, which are equal to −0.5 and 4.5, respectively.

A continuous piecewise function, used for calculating the final output score (*y*) from $y_{rd}$, is expressed as
(4)$$y=\left\{\begin{array}{c} \frac{1}{p_{1}-b_{l}}y_{rd}-\frac{b_{l}}{p_{1}-b_{l}}+b_{l},b_{l}<y_{rd}\leq p_{1}\\ \frac{1}{p_{2}-p_{1}}y_{rd}-\frac{p_{1}}{p_{2}-p_{1}}+b_{l}+1,p_{1}<y_{rd}\leq p_{2}\\ \frac{1}{p_{3}-p_{2}}y_{rd}-\frac{p_{2}}{p_{3}-p_{2}}+b_{l}+2,p_{2}<y_{rd}\leq p_{3}\\ \begin{array}{c} \frac{1}{p_{4}-p_{3}}y_{rd}-\frac{p_{3}}{p_{4}-p_{3}}+b_{l}+3,p_{3}<y_{rd}\leq p_{4}\\ \frac{1}{b_{u}-p_{4}}y_{rd}-\frac{p_{4}}{b_{u}-p_{4}}+b_{l}+4,p_{4}<y_{rd}<b_{u} \end{array} \end{array}\right.$$
where $p_{1}$, $p_{2}$, $p_{3}$, and $p_{4}$ are the values of $y_{rd}$ at the four demarcation hyperplanes, respectively, and can be expressed as
(5)$$p_{i}=\frac{5}{1+e^{-\frac{D_{i}}{\left||W|\right|}+2}}-0.5,i=1,2,3,4$$

The five segments of (4) correspond to the five levels of gait task in UPDRS. With (4), y are valued from 0.5, 1.5, 2.5, and 3.5 at the four demarcation hyperplanes, respectively.

Considering the practical situation, constraints are added in the proposed model as
(6)$$\left\{\begin{array}{c} p_{1}-b_{l}\geq 0\\ p_{i+1}-p_{i}>0,i=1,2,3\\ b_{u}-p_{4}\geq 0 \end{array}\right.$$

To further illustrate the proposed model and the relationships between $x_{hp}$, $y_{rd}$, and y, we present an example in Figure 2.

#### 2.5.2. Training and Validation

During each walking trial, gait features were collected from ten walking steps, and mean values of gait features were calculated and used later. In total, 53 data points were collected from the 53 participants.

To avoid overfitting, the redundant features need to be removed. For this purpose, one-way analysis of variance (ANOVA) is used for preliminary selection of the features. The feature which shows little significant statistical difference (*p* < 0.01) between different subjects would be removed in this operation.

For simplifying the training process, the coefficients w, c, and $p_{1}$ to $p_{4}$ in the proposed model are trained, instead of W, C, and $D_{1}$ to $D_{4}$. For training the model, the fmincon library of MATLAB is applied, which is used to find the optimal coefficients that satisfy the minimum loss function (L). The parameters are defined as
(7)$$E_{j}=\frac{\sum_{i=1}^{n_{j}}|y_{j,i}-Y_{j,i}|}{n_{j}}$$
(8)$$R_{1}=\sum_{k=1}^{M}(w_{k}•\mathrm{std}\left(x_{k}\right){)}^{2}$$
(9)$$R_{2}=\frac{1}{b_{u}-p_{4}}+\frac{1}{p_{4}-p_{3}}+\frac{1}{p_{3}-p_{2}}+\frac{1}{p_{2}-p_{1}}+\frac{1}{p_{1}-b_{l}}$$
(10)$$L=\left(\sum_{j=0}^{N}E_{j}\right)/N+\lambda R_{1}+\beta R_{2}$$
where *i*, *j* represents the *i*th data point that is rated as level *j* by neurologists, $n_{j}$ is the number of data points under level *j*, *y* is the score estimated by the propose model, *Y* is the level rated by neurologists, *M* is the number of used gait features, *N* is the number of levels appeared in the training dataset, $E_{j}$ is the mean absolute estimation error under level *j*, $R_{1}$ is a $L_{2}$ regularization term for preventing overfitting, and $R_{2}$ is a compensation term for preventing too close between hyperplanes. std($x_{k}$) is the standard deviation of the set of the *k*th gait feature in the training dataset, $w_{k}$ is the corresponding coefficient of the *k*th parameter in the model, λ and β are the weight of $R_{1}$ and $R_{2}$, respectively, and the λ and β are selected as 0.1 and 0.005 in this study.

The initial value of w and c in the trained linear part are set as zero. The initial values of $p_{1}$ to $p_{4}$ are set as 0.5, 1.5, 2.5, and 3.5, respectively.

Leave-one-subject-out (LOSO) cross-validation [28] is performed. This implies that when validating the performance of the proposed model on each participant, the data points from the rest of the 43 participants are used for training the model, and the data point of this participant is used for testing the model.

### 2.6. Traditional Models for Comparisons

Classifiers and linear regression models have been used in many existing studies for automatically rating the levels of gait task in UPDRS. For comparison purposes, a SVM classifier, a naive Bayes classifier, and a multiple linear regression (MLR) model, which are widely used by current researchers for automatic classification, are tested by the same dataset. LOSO cross validation is also implemented on the three traditional models.

### 2.7. Statistical Analysis

For calculating the estimation errors of the models, the estimated scores are rounded into integers as the form of five levels of gait task in UPDRS according to their values. The estimation error (e) is calculated as
(11)$$e=|s_{e}-s_{n}|$$
where $s_{e}$ is the rounded score and $s_{n}$ is the level rated by neurologists. Cumulative distributions ($c_{i}$) of e are also calculated, as
(12)$$c_{i}=\frac{n_{e\leq i}}{N_{T}}•100\%$$
where $n_{e\leq i}$ is the number of estimated scores with an estimation error less than or equal to *i* and $N_{T}$ is the total number of the estimated scores.

Recursive feature elimination (RFE) is implemented for each model to investigate the relationship between the number of features and the accuracy and is also used to find the optimal set of features. Starting with an empty feature set and ending until all the features are involved, the REF would add certain features, which provide maximum decrease or minimum increase on the mean estimation error [29]. The accuracy obtained during this process is recorded.

## 3. Results

### 3.1. Gait Features Distributions

Figure 3 shows the distributions of each gait feature under different gait task levels among the PD participants. For the convenience on presentation and comparison, the gait features have been standardized by the mean values and standard deviations obtained from healthy participants. The range of healthy participants are also plotted in this figure. One-way ANOVA is used for the preliminary selection of the features. The *p*-values of SL, MH, RSZ, MPV, MSV, and MHD are less than 0.01, which indicates the significance of the distribution variance between different gait task levels. The other six gait features are removed from the feature set as redundant information.

### 3.2. Results of Gait Task Assessment

Figure 4 shows the results of RFE for the proposed nonlinear model and the three traditional models. Obviously, for different learning models, the number and types of optimal feature combinations are different, and blindly increasing the number of features will even increase the error of the system, which are consistent with the statement that neuroscientists who evaluate Parkinson’s gait always focus on a few gait features. The optimal feature set for the proposed nonlinear model consists of feature MH, SL, and RSZ. The mean value of the gait task level estimation error calculated by (11) is 0.151. The optimal feature set for the SVM model consists of MSV, SL, and RSZ, with a mean estimation error of 0.264. The optimal feature set for the naive Bayes model consists of SL and RSZ with a mean estimation error of 0.264. The optimal feature set for the linear regression model consists of MH, RSZ, and MHD with a mean estimation error of 0.340.

Figure 5 shows the estimation error distributions of the four models with their own optimal feature set. In order to more intuitively evaluate and compare the automatic scoring results of the four models, we mainly focus on two scenarios, namely accurate scoring (e = 0) and slight deviation (e ≤ 1). In terms of percentages of participants, 84.9%, 73.6%, 73.6%, and 66.0% of the participants are accurately assigned into the true level with the proposed nonlinear model, the SVM model, the naive Bayes model, and the linear regression model, respectively. Regarding estimation error, 96.2%, 92.5%, 96.2%, and 98.1% of the participants have an estimation error not more than 1 with the proposed nonlinear model, the SVM model, the naive Bayes model, and the linear regression model, respectively.

Figure 6 shows a boxplot of the distribution of the gait task scores estimated by the proposed nonlinear model with the optimal feature set. The theoretical score range of each gait task level is also marked in the figure. For the healthy subjects, the estimated gait task scores are 0.26 ± 0.32 (mean ± standard deviation). The estimated gait task scores are 0.31 ± 0.12, 1.27 ± 0.70, and 1.64 ± 0.66 for PD subjects in level 0, 1, and 2, respectively.

Figure 7 shows the accuracy of different methods to score HC and PD. Among them, the nonlinear model proposed in this paper has 2, 0, 3, and 3 errors for HC and PD with a level of 0 to 2, respectively. The error of SVM is 3, 0, 3, and 8, and the error of naive Bayes is 3, 1, 2, and 8, respectively. For the linear model, the errors are 4, 3, 1, and 10, respectively.

## 4. Discussion

In this study, a novel nonlinear model is developed for automatically classifying the levels of gait task in UPDRS. According to the results shown in Figure 5, 84.9% of the subjects are accurately rated as their actual levels, and the results are significantly better than other traditional algorithms (73.6%, 73.6%, and 66.0% for the support vector machine model, the naive Bayes model, and the linear regression model, respectively). In addition, according to the results shown in Figure 6, the distribution of the estimated scores of PD patients in level 0 is similar with that of the healthy subjects, which matches the UPDRS definition of level 0 in gait task. The estimated scores of PD patients show an obvious positive correlation with the actual levels. These results also indicate good performance of the proposed method.

To accomplish the comparisons of the different models for the automatic scale assessments, the two classifier models and a linear regression model are also performed, which have been used in previous studies. According to the comparison results shown in Figure 4, Figure 5, and Figure 7, the proposed nonlinear model is generally superior to the other three methods but not all aspects, as linear models and naive Bayes are more superior for subjects rated level 1 (only 1 subject with level 1 was mis-scored, while the proposed model had 3 subjects with level 1 mis-scored). However, from practical clinical considerations, we are more faced with the scoring of patients from level 0, 1, and 2 rather than limited to level 1 only. Therefore, the nonlinear model proposed in this paper has more application value. Among the four methods, the linear model appears to have the largest error which is caused by the hardness for linear model to match the nonlinearities existing in the rating process. The decision boundaries in the proposed model for different levels are parallel with each other while in the traditional classifier models are not. The parallel decision boundaries is helpful for further quantifying the extent of gait abnormality of a data point withn its level, which enables the proposed model to output the detailed score but not just the level. With the proposed model, the data points in the same level can also be compared. This would make sense in the evaluation of rehabilitation and treatment effectiveness for patients in short term.

A RFE method is implemented in our study for finding the optimal feature set. As Figure 4 shows, to reach the highest accuracy, only a small set of features is needed. The similar phenomenon also happened in some previous studies [29]. This is reasonable because the neurologists only focus on a few main aspects of gait which are special in PD patients during the assessment process. The low-dimensional gait features are sufficient to accomplish the assessment, which can reduce the complexity of model and improve generalization ability.

Only two shank-mounted IMUs are involved in the proposed system. It has a great potential to be applied in the daily environment for long-term assessment of PD gait due to its convenience and comfort [30]. The long-term assessment can monitor the variance of gait in few days to few months, which is meaningful for the evaluation of the progression of the disease and the effectiveness of treatments.

There are some limitations in this study. First, the rating of gait task by neurologists and the collection of IMU data were not at the same time in one day, which may cause some inevitable errors between calculated and actual values. Second, because most PD patients recruited in our study are in middle or late stage, the data are not collected evenly. Most PD subjects are in level 2 of gait task in UPDRS. Only a few PD subjects are in level 0 and level 1. The overall number of PD subjects is also smaller than some previous studies [5]. Overfitting is prone to occur due to the small and unevenly distributed dataset, brings additional errors, and reduces the practicability. In addition, although the proposed gait features show a good performance in the automatic assessment of gait scores, the selection of them still deserves further investigating. Only limited gait features are extracted from the motion of shank and ankle. Other features commonly used in literature for gait classification or evaluation such as the variance of gait parameters [15] are ignored in this study. These features would be considered in future work to improve the current accuracy of the proposed model.

## 5. Conclusions

In this paper, a novel method on automatic assessment of the gait task in UPDRS is developed based on only two shank-mounted IMUs and 12m straight walking test. With the proposed nonlinear model and under the optimal feature set, 84.9% of the recruited subjects were accurately rated as their actual gait task levels, and 96.2% of the subjects have a level rating error not more than 1. This indicates the good conformance between the automatic rating results and the results from neurologists. For comparison purpose, several traditional models were tested by the same dataset. Compared with the results of other three models that 73.6%, 73.6%, and 66.0% of the recruited subjects were accurately rated as their actual gait task levels by the support vector machine model, the naive Bayes model, and the linear regression model, the proposed nonlinear model shows a much higher accuracy because of the well matching with the nonlinearities existing in the assessment. However, for safety reasons, this method is not suitable for patients with level 3 and 4 of UDRRS. The proposed model can further quantify the extent of gait abnormality of a data point within its level, which enable the proposed model to output the detailed score but not just the level. This is helpful for the evaluation of rehabilitation and treatment effectiveness in short term. The approach has great potential for further clinical applications as a convenient and objective tool on automatically classifying the levels of gait task in UPDRS. 

## Figures and Tables

**Figure 1 sensors-23-02104-f001:**
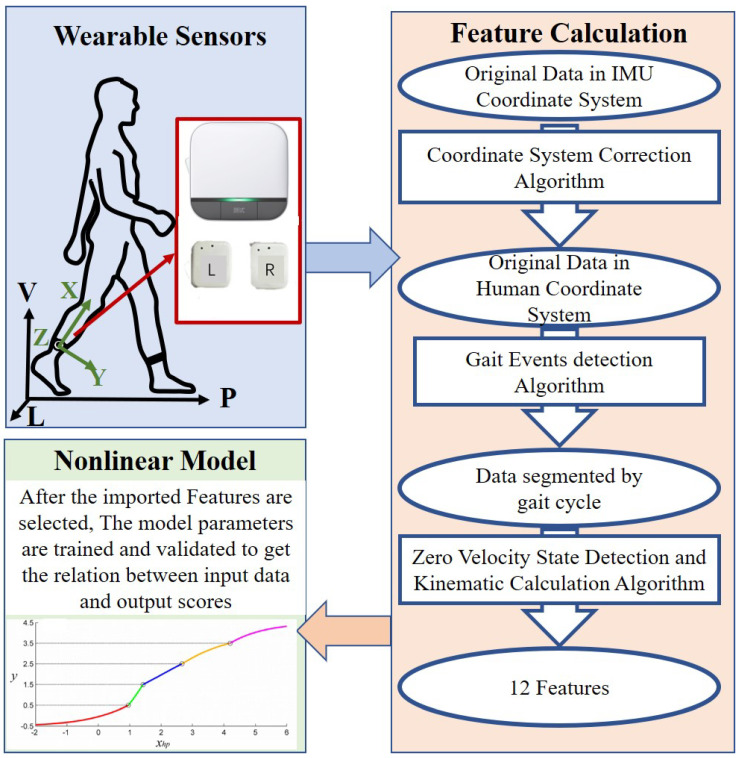
The configuration of the wearable sensors and actuators system.

**Figure 2 sensors-23-02104-f002:**
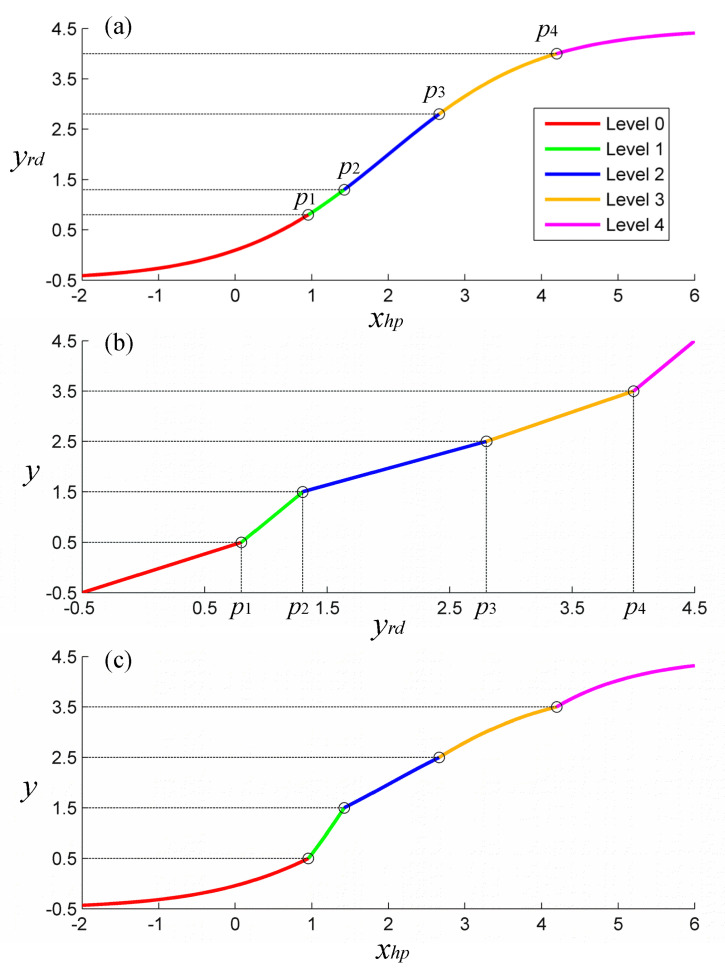
An example of the proposed model. The segments of the curve correspond to the five levels of gait task in UPDRS (0 to 4). (**a**) The $x_{hp}$ – $y_{rd}$ plot. (**b**) The $y_{rd}$ – y plot. (**c**) The $x_{hp}$ – y plot.

**Figure 3 sensors-23-02104-f003:**
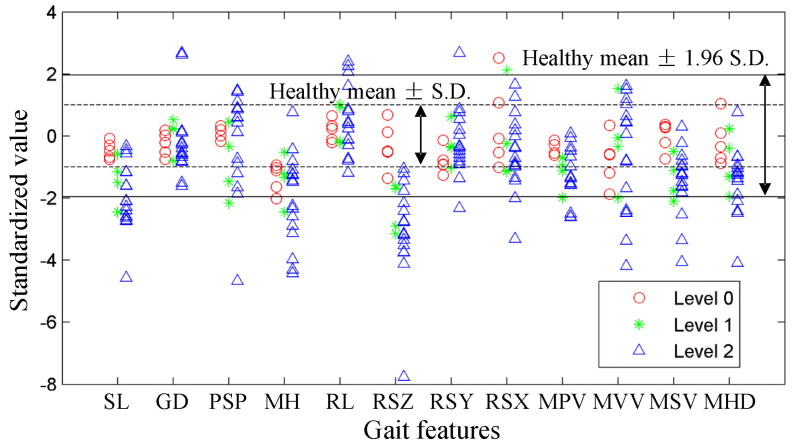
The distributions of the gait features under different gait task levels among the PD participants. Each point represents one data point. The gait features have been standardized by their mean values and standard deviations of the healthy participants.

**Figure 4 sensors-23-02104-f004:**
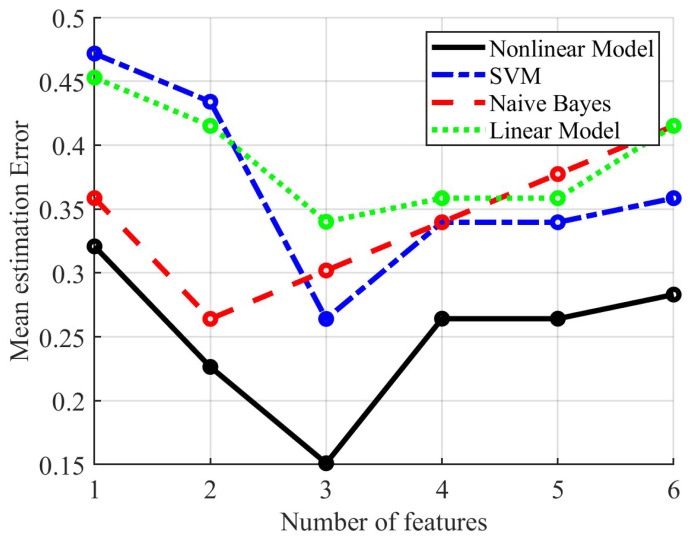
The RFE results of the proposed nonlinear model and the three traditional models.

**Figure 5 sensors-23-02104-f005:**
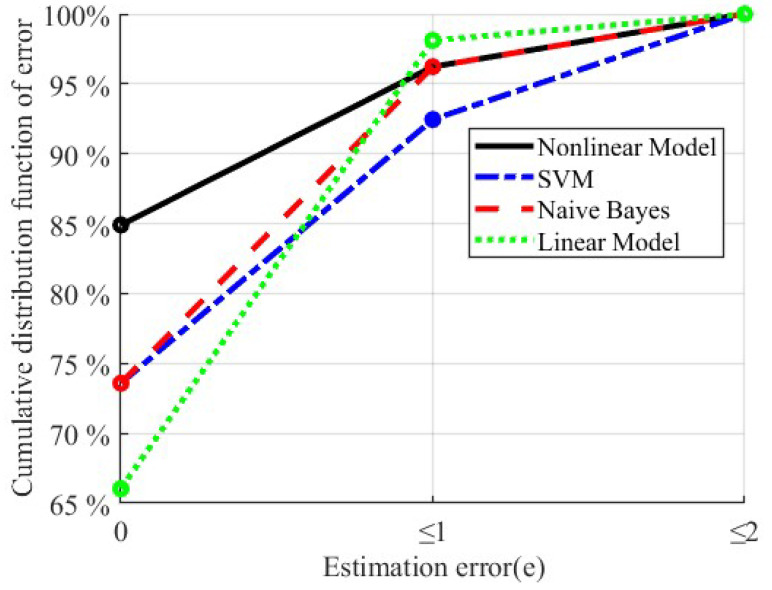
The cumulative distribution function of estimation error for the proposed nonlinear model and the three traditional models.

**Figure 6 sensors-23-02104-f006:**
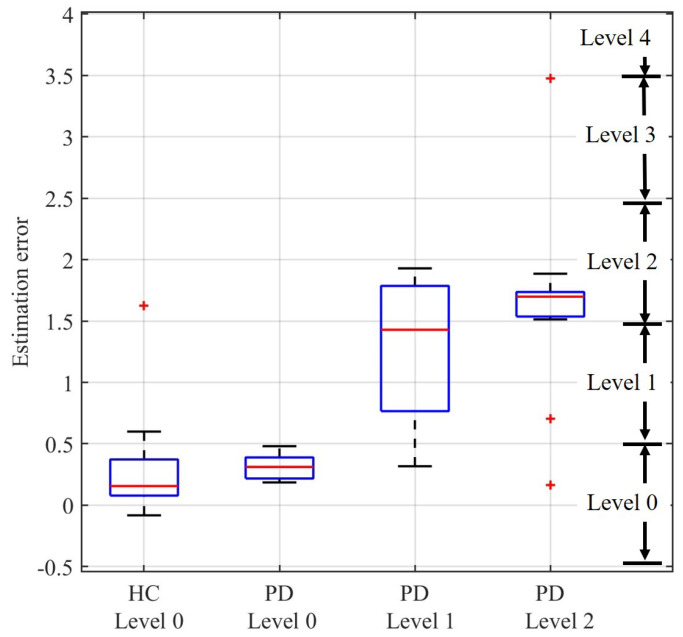
The distribution of the gait task scores estimated by the proposed nonlinear model. HC and PD represent the healthy control subjects and the subjects with Parkinson’s disease, respectively. The theoretical score range of each gait task level is marked in the figure. The box presents the 50% of the estimated scores around the median line (namely the red horizontal segment). Its vertical width gives a direct indication of the value of the interquartile range. The whiskers indicate the maximum and minimum value of the scores. The outliers (+) are also plotted.

**Figure 7 sensors-23-02104-f007:**
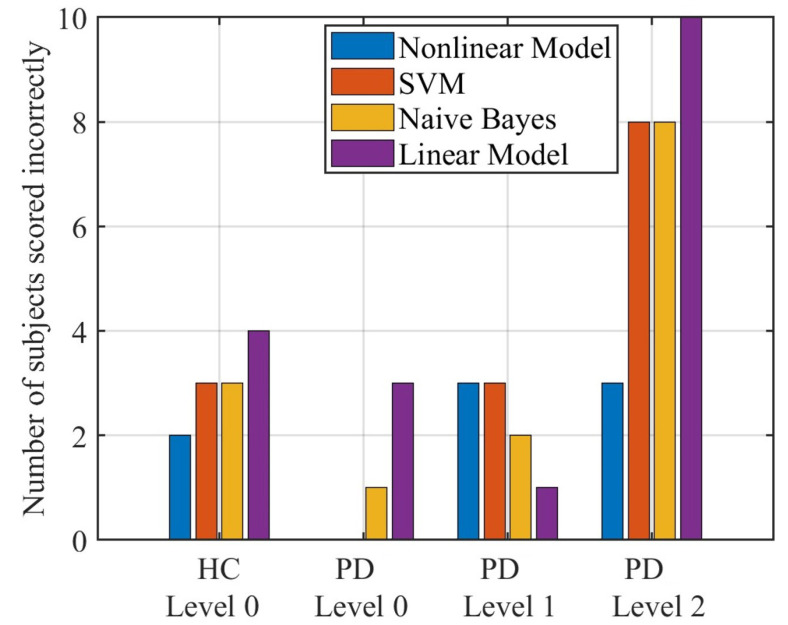
The number of HC and PD scored incorrectly by different methods.

**Table 1 sensors-23-02104-t001:** The information of subjects.

Subjects	UPDRS	Height/m	Weight/kg	Age/Years	Number of Subjects
	0				5
Subjects with	1	1.64 ± 1.07	62 ± 8	69 ± 8	4
Parkinson	2				16
Young Healthy subjects	0	1.67 ± 1.24	62 ± 11	32 ± 5	19
Old Healthy subjects	0	1.58 ± 3.13	56 ± 6	69 ± 7	9

**Table 2 sensors-23-02104-t002:** The definitions of gait parameters.

Gait Parameter	Definition	RMSE (Validated in [24])
Stride length (SL)	The height-normalized linear displacement between two adjacent ankle landing points.	2.3 m
Gait cycle duration (GD)	The duration between two adjacent HS events.	14 ms
Percentage swing phase (PSP)	The duration between adjacent TO event and HS event divided by GD.	no validated
Max ankle height (MH)	The maximum value of height-normalized ankle displacement in V-axis direction during the gait cycle.	1.0 cm
Range of lateral displacement (RL)	The range of height-normalized displacement along the L-axis direction during the gait cycle.	no validated
Range of shank Z-axis rotation (RSZ)	The range of the integration of IMU Z-axis angular velocity during the gait cycle.	no validated
Range of shank Y-axis rotation (RSY)	The range of the integration of IMU Y-axis angular velocity during the gait cycle.	no validated
Range of shank X-axis rotation (RSX)	The range of the integration of IMU X-axis angular velocity during the gait cycle.	no validated
Max ankle progressive velocity (MPV)	The maximum value of height-normalized ankle velocity in P-axis direction during the gait cycle.	3.0 cm/s
Max ankle vertical velocity (MVV)	The maximum value of height-normalized ankle velocity in V-axis direction during the gait cycle.	no validated
Max shank Z-axis angular velocity (MSV)	The maximum value of IMU z-axis angular velocity during the swing phase.	no validated
Ankle displacement at MH (MHD)	The height-normalized ankle displacement in P-axis direction when MH occurs.	1.9 cm

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
