# Peer review of "Automatic Assessments of Parkinsonian Gait with Wearable Sensors for Human Assistive Systems"

_sensors, 2023, doi:10.3390/s23042104_

Round 1

Reviewer 1 Report

Abstract: Improve the abstract. Be sure that the problem to solve, the objective, methods, results and conclusions are in the abstract. The results should be quantitative, What is a “good performance” , how good is the new model compared with the other three traditional models?

The first two paragraph are almost identically, please revise the wording.

Specific notes:
Line 38-39: Revise the sentence.

Line 72: Body Sensor instead of Boby Sensor

Line 80: What are the actual clinical needs? Describe them, quantitatively if possible.

Line: 86-87: Refer to table 1, eliminate the part “of all ages”

Line 93: What are the accuracies of the other models?  
From Line 85 to 97 Could be easily described in the abstract, to show to the readers the advantages and improvements of your proposal.

Line 104: Include the name of the hospital and department, Include the time span of the measures. ¿When did you start the experimentation, when it end it?

Table 1: Why did you used healthy subjects with so wide range of ages? It is widely know that the gait changes with age, you should used subjects with ages similar to your Parkinson´s Disease group for a better comparison. Please explain your reasons.
Describe better the range of ages of the healthy subjects. How their ages are distributed?

Sex of the subjects? How many female, how many males?

Line 124: What are the weight, sampling rate and size of the devices? How were they attached to guarantee and stable connection to the measurement site in order to avoid unwanted movement artifacts?

Figure 1: The image of the Exoskeleton and your previous comments make me think that your work includes the exoskeleton control. I mean, that your subjects were wearing the exoskeleton while making the test and that your system corrected the actuators. But by logic, I guess it is not the case. Please be clear on your set-up and avoid data and images that could lead to misunderstandings. In the Linear Model text-box, they are mentioned w, c, p1 and p4, which were not previously mentioned. Please be sure they are mentioned previously or mention them in the figure's title.

Line 122: Here you mention the sampling rate, be sure that technical characteristics of the devices are described in the same paragraph. How did you synchronized the devices?

Table 2: Please, include in the table the following information: The reference on which the calculations are based. I some reader wants to replicate your work, what reference should he/her to use to calculate that specific parameter?. Also, for those parameters were you have the data, write the precision. It is clearer and better to add this information on the table instead of put it in the text as you did.

Line 164: Please give more details about the normalization of the data.

Line 167+5: It is not clear what you mean about the “actual range”. 0±0.5? IT means that your range could be -0.5? If the UPDRS model only uses values starting from zero, what means -0.5? If the output is continuous, what is ±0.5? Please explain better.
 Equation 2: What does lowercase omega mean? Is the same as “w”? Is because of the style of the fonts? Please be sure that the typography in equations and text is the same.

Line after the equation 1: This C, and C are those referred to in Fig. 1? If so, use the same type of style for the letters (uppercase or lowercase)

What is the idea behind the model? Is it based on a previous model? Why did you select a sigmoid function? Please describe better the line of thoughts to develop your model.

Line 178: I supposed that none of the subjects were using an exoskeleton, right?

Line 202: Please cite the references where these models have been used, and the obtained results. I suggest using a table to show works in which those models were used, the number of patients and the results.

Lines 206+3: Who were the neurologist? How many? Years of experience? Describe those data in the Method section.

Please: Use the appropriate verb tenses (past, present and future). Use the past tense to report what happened in the past: what you did, what someone reported, what happened in an experiment, and so on. It is annoying to read verbs in present to explain tasks that you have already done during the development of your work. Form example: “During each walking trial, gait features ARE (were) collected from ten walking steps and mean 178 values of gait features ARE (were) calculated and used later. 53 data points ARE (were) totally collected 179 from the 53 participants.”

Figure 3: The values presented are only from the PD patients? Or do they also include the healthy subjects?

Please describe the assumptions and characteristics for implementing the other models (SVM, Naïve Bayes and Linear model)

Line 251: Mention what are the values reached by the other models.

Line 261, explain quantitatively, how much is superior to the Naïve Bayes method compared to your model and the others.

Line 263: What is the opinion of the neurologist for the new proposed continuous scale, compared with the discrete scale (1, 2, 3, 4)

Line 266, Caued- caused

Line 286: I suggest not to mention here the experiment with the exoskeleton, it only causes confusion, But if you want to add it, explain better what did you do, the results (quantitatively), etc.  

Finally, improve conclusions. Add more quantitative data regarding the comparison between your model and the other three. State clearly that your model was tested only in patients with a level form zero to two in the UPDRS scale, what it requires to be implemented (sensors, and tasks).

Author Response

Abstract: Improve the abstract. Be sure that the problem to solve, the objective, methods, results and conclusions are in the abstract. The results should be quantitative, What is a “good performance” , how good is the new model compared with the other three traditional models?

Thank you for your comments, a digital presentation of the results does provide a visual illustration of the significance of the manuscript. We have updated the manuscript by including a detailed description of the results in the Abstract.

The first two paragraph are almost identically, please revise the wording.

 We have updated the manuscript by simplifying the description in the first paragraph to make the logical relationship between the first two paragraphs clearer: the first paragraph mainly talks about the importance and limitations of UPDRS in the evaluation of gait task in Parkinson's disease. The second paragraph describes the irreplaceable role of IMU in the quantitative evaluation of Parkinson's gait compared with other instruments.

Specific notes:
Line 38-39: Revise the sentence.

We have corrected the sentence in Line 31

Line 72: Body Sensor instead of Boby Sensor

We have corrected the word.

Line 80: What are the actual clinical needs? Describe them, quantitatively if possible.

Thanks for your comments, the actual clinical need here refers to the proposed method being able to perform automatic scoring of gait tasks with clinically acceptable accuracy based on a lightweight wearable system. Regarding "clinically acceptable accuracy", there are no specific clinical regulations at present, but from the perspective of Line 37-38, the accuracy needs to be at least greater than the accuracy of manual classification in [14].

Line: 86-87: Refer to table 1, eliminate the part “of all ages”

We have corrected it in Line78-79

Line 93: What are the accuracies of the other models?  

We have addressed the problem in Line 84-85
From Line 85 to 97 Could be easily described in the abstract, to show to the readers the advantages and improvements of your proposal.

We have updated the contents, to make it clear to understand for readers.

Line 104: Include the name of the hospital and department, Include the time span of the measures. ¿When did you start the experimentation, when it end it?

We have updated it in Line 95-96

Table 1: Why did you used healthy subjects with so wide range of ages? It is widely know that the gait changes with age, you should used subjects with ages similar to your Parkinson´s Disease group for a better comparison. Please explain your reasons.
Describe better the range of ages of the healthy subjects. How their ages are distributed?

The reason why we choose multiple age groups is mainly for the following two considerations: 1. Younger subjects are easier to collect and abundant samples can ensure the robustness of the algorithm 2. Compared with the elderly, young people rarely suffer from other neurological diseases or chronic diseases, so unreasonable interference can be eliminated and the purest healthy gait data can be obtained.

A more detailed distribution of the age of healthy subjects has been updated in Table 1 and Line78-79

Sex of the subjects? How many female, how many males?
Thank you for your question, the gender information of the subjects was not collected in this experiment, because we think it is irrelevant to gait performance

Line 124: What are the weight, sampling rate and size of the devices? How were they attached to guarantee and stable connection to the measurement site in order to avoid unwanted movement artifacts?

The size of the instrument is only 25*40*15mm and the mass is 100g. which is very light and compact, so it will not become a burden for the subjects to walk. And the data is sampled at a frequency of 100Hz. The instrument is attached by elastic straps to the outer side of the shank near the ankle, where there are very few muscles, so there is little possibility of slippage during walking.

We also emphasize this point in the Line 117 of the latest manuscript

Figure 1: The image of the Exoskeleton and your previous comments make me think that your work includes the exoskeleton control. I mean, that your subjects were wearing the exoskeleton while making the test and that your system corrected the actuators. But by logic, I guess it is not the case. Please be clear on your set-up and avoid data and images that could lead to misunderstandings. In the Linear Model text-box, they are mentioned w, c, p1 and p4, which were not previously mentioned. Please be sure they are mentioned previously or mention them in the figure's title.

Thanks for your suggestion, we have updated Figure 1

Line 132: Here you mention the sampling rate, be sure that technical characteristics of the devices are described in the same paragraph. How did you synchronized the devices?
We added a description of the sampling rate in Line 117,As shown in Figure 1, this device consists of a host and two slaves. The host communicates with the slave via Bluetooth. During measurement, the host sends data collection instructions to two slaves at the same time, and the slaves receive the instructions and start collecting data to ensure data synchronization

Table 2: Please, include in the table the following information: The reference on which the calculations are based. I some reader wants to replicate your work, what reference should he/her to use to calculate that specific parameter?. Also, for those parameters were you have the data, write the precision. It is clearer and better to add this information on the table instead of put it in the text as you did.

Thank you for your suggestion. The gait parameters in the table 2 are generally accepted and commonly used by researchers at present, and they all have corresponding clear definitions and calculation method. Considering the limited space, we will not repeat them in this article. The accuracy display you mentioned is very valuable, which we have updated in table 2 of the latest manuscript 

Line 164: Please give more details about the normalization of the data.

These parameters are all length-related parameters, that is, vary with the height of the subjects, so we divide these parameters by their height

Line 167+5: It is not clear what you mean about the “actual range”. 0±0.5? IT means that your range could be -0.5? If the UPDRS model only uses values starting from zero, what means -0.5? If the output is continuous, what is ±0.5? Please explain better.

All our results here are continuous, just like -0.5 to +0.5 belong to the 0 level in UPDRS in our method, but in order to solve the limitations of UPDRS itself, we decided to use continuous, that is- 0.5 represents the most healthy performance in level 0 of UPDRS, and +0.5 represents the worst performance in level 0 of UPDRS, so it can be used to help monitor the patients’ rehabilitation process with the same level.

Equation 2: What does lowercase omega mean? Is the same as “w”? Is because of the style of the fonts? Please be sure that the typography in equations and text is the same.

Thank you for your comment, it is indeed the meaning of ‘w’, and we have corrected it in the latest manuscript

Line after the equation 1: This C, and C are those referred to in Fig. 1? If so, use the same type of style for the letters (uppercase or lowercase)

This is a error in the drawing of the Figure 1, we have corrected it, c has no meaning in Figure 1

What is the idea behind the model? Is it based on a previous model? Why did you select a sigmoid function? Please describe better the line of thoughts to develop your model.

The idea of our model is inspired by previous work. We found that the previous automatic scoring of Parkinson's level based on linear models often had poor results. Therefore, we decided to use nonlinearity. The sigmoid function is mainly to limit the value range of the function output while ensuring the results is smooth and flat

Line 178: I supposed that none of the subjects were using an exoskeleton, right?

At present, the completed work of the manuscript is based on natural walking, as explained in 2.1 Participants and Protocols. Combining sensing results with exoskeleton items is more future work, which is mentioned in the discussion.

Line 202: Please cite the references where these models have been used, and the obtained results. I suggest using a table to show works in which those models were used, the number of patients and the results.

Thank you for your comments, the expression here may cause some misunderstanding, which has been corrected. what we want to express here is that classifiers and linear regression models are commonly used for automatic scoring of Parkinson's patients, so we chose the most common classifiers to compare with our proposed method. These classifiers are not all used by previous scholars for Parkinson's automatic scoring (the related work is really very little), so if it is made into a table, other unrelated work may be introduced.

Lines 206+3: Who were the neurologist? How many? Years of experience? Describe those data in the Method section.

This issue is very important, we have updated it on Line97-98

Please: Use the appropriate verb tenses (past, present and future). Use the past tense to report what happened in the past: what you did, what someone reported, what happened in an experiment, and so on. It is annoying to read verbs in present to explain tasks that you have already done during the development of your work. For example: “During each walking trial, gait features ARE (were) collected from ten walking steps and mean 178 values of gait features ARE (were) calculated and used later. 53 data points ARE (were) totally collected 179 from the 53 participants.”

 Thank you a lot, we have revised the verb tenses of the full manuscript, and hope that this modification can better meet the requirements of the journal requirements

Figure 3: The values presented are only from the PD patients? Or do they also include the healthy subjects?

 As the figure title displays, Figure 3 shows the distribution results of Parkinson's patients at different levels, without including the data of healthy subjects, so as not to affect the display effect due to too many graphic elements

Please describe the assumptions and characteristics for implementing the other models (SVM, Naïve Bayes and Linear model)

All calculations are based on MATLAB2021a as mentioned in the manuscript, so the settings and characteristics of all traditional classifiers are based on the default settings of matlab's own toolbox commands. For example, when using SVM training, only use svmStruct_trained = fitcsvm (data_feature, datalabel) to carry out, which is why the three classifier settings are not elaborated in the manuscript.

Line 251: Mention what are the values reached by the other models.

We have updated it in Line241-242 of 4. Discussion

Line 261, explain quantitatively, how much is superior to the Naïve Bayes method compared to your model and the others.

We have updated it in Line253-254 of 4. Discussion

Line 263: What is the opinion of the neurologist for the new proposed continuous scale, compared with the discrete scale (1, 2, 3, 4)

At present, the article is still in the stage of theoretical discussion and has not yet been practically applied in clinical application.

Line 266, Caued- caused

We have corrected it.

Line 286: I suggest not to mention here the experiment with the exoskeleton, it only causes confusion, But if you want to add it, explain better what did you do, the results (quantitatively), etc.  
We have removed contents concerning exoskeletons, which does make the structure of the article clearer

Finally, improve conclusions. Add more quantitative data regarding the comparison between your model and the other three. State clearly that your model was tested only in patients with a level form zero to two in the UPDRS scale, what it requires to be implemented (sensors, and tasks).

Thanks for your comments, we have updated the conclusion, containing the results, required instruments and tasks.

Reviewer 2 Report

Dear authors

this is nicely conducted study. author is trying to create more objective instrument to assess subjective symptoms. assessment will be more objective and it will make assess of therapy more injective.

do you have updrs motor part 3 scores to compare with you assessment. That will also validate severity associated changes assessed by your instrument. 

please reduce size of introduction. 

Author Response

Thanks for your comments, the response is as follows.

do you have updrs motor part 3 scores to compare with you assessment. That will also validate severity associated changes assessed by your instrument. 
Thank you very much for your suggestion, but level 3 patients do not have the ability to walk independently, and need corresponding assistive equipment to complete the walking task. Therefore, for the sake of experimental safety, level 3 patients are not included in the scope of our subject recruitment

please reduce size of introduction. 

Thanks for your comments, we have updated the introduction.The logical order of the present introduction begins by emphasizing that gait assessment is an important part of Parkinson's disease assessment. At present, UPDRS is commonly used in clinic to complete this work, but the results are often interfered by subjective factors. Next, analyze and compare the current instruments used to quantify human gait characteristics, highlighting the advantages of IMU. Then, we analyze and discuss the current researches based on the combination of IMU data and clinical evaluation system. It is discussed that the current part of the work is lacking in the combination with UPDRS. Therefore, the significance of this work is drawn.

Reviewer 3 Report

This manuscript proposes a method for the automatic assessment of gait tasks in UPDRS using two shank-mounted IMUs. The proposed nonlinear model showed relatively good conformance with the results from neurologists. The authors also discuss potential clinical applications of the method.

Overall, the proposed dialogue system is good to some extent but there are some doubts about its details of the experiment, model, and future application. However, there are some major problems to be clarified, which are listed below. As a result, the Reviewer believes that the paper cannot be accepted in its current form, and major revision is required.

(1)    The distribution of the selected participants with PD, classified as level 0-2, is not evenly distributed, and the age of these participants is concentrated in the elderly stage. In contrast, the healthy participants are mainly in the middle-aged stage. It would be beneficial for the authors to provide an explanation for this discrepancy and its potential impact on the study's results.

(2)    Gait parameters are numerous, and it would be beneficial for the authors to provide a brief explanation of the rationale behind the selection of the parameters used in the study. For example, the authors could explain how certain medical symptoms may affect specific parameters, or how the chosen parameters have a higher correlation with the disease severity level. This would contribute to a deeper understanding of the study's results.

(3)    Some figures describing the model's performance lack an adequate explanation. Additionally, there may be occasional omissions in the figures that should be addressed. As an example, the y-axis label of Figure 4 may be missing the word "error", which would be beneficial for the reader's understanding. Furthermore, several figures appear blurry, and it is suggested that the authors ensure that the figures are of high resolution, so that the details are clearly visible and the information conveyed by the figures is easily understood by the readers.

(4)    The proposed model concept is similar to that of SVM, as both utilize a hyperplane to separate different classes of data. It is recommended that the authors provide a more detailed explanation of the distinctions between the proposed model and SVM, as well as the advantages of the new model in addressing the problem in question. This would further aid in the understanding and evaluation of the proposed method.

(5)    The proposed method in this manuscript utilizes IMU for automatic detection and has the potential for future application with exoskeleton devices for the ankle joint. However, the study only employs IMU sensors and it is suggested that the authors consider the potential effects of incorporating wearable exoskeleton devices on the gait of the wearer and how it may impact the parameters of gait and the overall effectiveness of the model. It is recommended that the authors provide a theoretical analysis or conduct further experimentation to evaluate the generalization of the proposed model to this new scenario.

Author Response

Thanks for your comments, the response is as follows.

(1)    The distribution of the selected participants with PD, classified as level 0-2, is not evenly distributed, and the age of these participants is concentrated in the elderly stage. In contrast, the healthy participants are mainly in the middle-aged stage. It would be beneficial for the authors to provide an explanation for this discrepancy and its potential impact on the study's results.

A more detailed distribution of the age of healthy subjects has been updated in Table 1 and Line78-79.

The subjects are composed of the elderly and young subjects. The role of the young subjects mainly has the following two reasons: 1. Younger subjects are easier to collect and abundant samples can ensure the robustness of the algorithm 2. Compared with the elderly, young people rarely suffer from other neurological diseases or chronic diseases, so unreasonable interference can be eliminated and the purest healthy gait data can be obtained.

(2)    Gait parameters are numerous, and it would be beneficial for the authors to provide a brief explanation of the rationale behind the selection of the parameters used in the study. For example, the authors could explain how certain medical symptoms may affect specific parameters, or how the chosen parameters have a higher correlation with the disease severity level. This would contribute to a deeper understanding of the study's results.

Your proposal is very meaningful, but in fact it is difficult to explain the clinical connection behind each specific gait parameter. For example, the gait characteristic of Parkinson's disease is often flustered gait. Obviously, stride length and gait cycle are the most relevant on the surface, but in practice, we found through data analysis that for different classification models, the weight of feature values is different. (see Line215-217)

(3)    Some figures describing the model's performance lack an adequate explanation. Additionally, there may be occasional omissions in the figures that should be addressed. As an example, the y-axis label of Figure 4 may be missing the word "error", which would be beneficial for the reader's understanding. Furthermore, several figures appear blurry, and it is suggested that the authors ensure that the figures are of high resolution, so that the details are clearly visible and the information conveyed by the figures is easily understood by the readers.

We have updated the Figure 2,4 and Results section to make readers clearer to understand.

(4)    The proposed model concept is similar to that of SVM, as both utilize a hyperplane to separate different classes of data. It is recommended that the authors provide a more detailed explanation of the distinctions between the proposed model and SVM, as well as the advantages of the new model in addressing the problem in question. This would further aid in the understanding and evaluation of the proposed method.

Thanks for your comments. A advantage of this model is that it can output continuous scoring data to achieve the evaluation of different disease degrees under the same level. We train the SVM model through the MATLAB2021 toolbox, which can only output classification results, not continuous values.

(5)    The proposed method in this manuscript utilizes IMU for automatic detection and has the potential for future application with exoskeleton devices for the ankle joint. However, the study only employs IMU sensors and it is suggested that the authors consider the potential effects of incorporating wearable exoskeleton devices on the gait of the wearer and how it may impact the parameters of gait and the overall effectiveness of the model. It is recommended that the authors provide a theoretical analysis or conduct further experimentation to evaluate the generalization of the proposed model to this new scenario.

Thank you for your comments. According to the suggestions of all reviewers, we have deleted the part of the paper that involves the application of exoskeletons (such as Figure 1, etc.), and the core of the current article is still in the sensing part of UPDRS automatic scoring

Reviewer 4 Report

The authors present a work on a wearable human assistive system to address automatic assessment of Parkinsonian gait. In general, this is an interesting work. However, there are many ways this work could be improved. I have  listed a few comments that would be useful to improve the quality of the paper.

  1. Even though this is a scientific paper, I have noticed informal words throughout the paper. One example is “rough” in the abstract, what do you mean? Not quantified at all or is it something else? I highly recommend getting rid of non-scientific words to explain results. It would be better to read the paper carefully.

  2.  The last sentence in the abstract “ The results indicate that the proposed method has good performance….. What does that mean? You need to quantify it ! In science we can not say good and bad.

  3. The main finding with qualitative/quantitative results are needed in the abstract.

  4. In 2.4: Feature calculation: ankle motion trajectories are discussed. It has been mentioned that to reduce the integrated error, zero velocity state of the ankle is detected. As far as I have experienced so far, a drift is involved when you take this approach. I would like to see how you compensated for it and report on the paper.

  5. 2.5.2 : Training and validation: What was the sampling frequency when you collected data from IMU? Please mention.

  6. Figure 2: Axis labelling or figure caption could be used to specify about axis, this is a minor comment.

  7. In discussion: It is mentioned that the results are significantly better than traditional methods: this is a huge ask,,, as if you would like to mention significant results, you need to report, also what are the traditional methods? You need to have a comparison.

  8. Is there any break for participants to minimise the fatigue? If that's the case, please report.

  9. Did you get any feedback from participants, especially from patients? I would suggest having a scale of 1 to 5 to mark how comfortable they were during the experiments, analyse your data and present them. 

  10. Conclusion: traditional methods are mentioned, what are they?

The current version of this paper is not ready for publication. However, a good attempt. 

Author Response

Thanks for your comments, the response is as follows

Even though this is a scientific paper, I have noticed informal words throughout the paper. One example is “rough” in the abstract, what do you mean? Not quantified at all or is it something else? I highly recommend getting rid of non-scientific words to explain results. It would be better to read the paper carefully.

Thank you very much for your comments, we have revised the wording of the full text, and hope to make the manuscript meet the needs of publication)

 The last sentence in the abstract “ The results indicate that the proposed method has good performance….. What does that mean? You need to quantify it ! In science we can not say good and bad.

Thank you for your comments, a digital presentation of the results does provide a visual illustration of the significance of the manuscript. We have updated the manuscript by including a detailed description of the results in the Abstract.

The main finding with qualitative/quantitative results are needed in the abstract.

That’s right, we have updated the abstract by offering the qualitative results

In 2.4: Feature calculation: ankle motion trajectories are discussed. It has been mentioned that to reduce the integrated error, zero velocity state of the ankle is detected. As far as I have experienced so far, a drift is involved when you take this approach. I would like to see how you compensated for it and report on the paper.

The details of the method can be found in reference 24- C. Gait Variable Selection and Estimation- 2) Detection of Ankle Zero-Velocity State, which is simply divided into two parts: detection and correction.

The detection part mainly uses that when the ankle joint is at zero speed, the speed of the IMU in the forward direction can theoretically be obtained by integrating the acceleration, or by multiplying the angular velocity of the IMU by the distance from the ankle joint to the IMU. Taking the speed difference obtained by these two methods, we consider the point with the smallest difference to be the zero-speed point of the ankle joint.

The correction part mainly uses the difference between the speed calculated by the ankle joint and the theoretical zero value when the ankle joint is at zero speed as the offset (marked as e, assuming), and the speed between the current zero speed moment and the previous zero speed moment could be corrected. During the stance phase during this period, we think that the offset is constant (due to the small displacement in the stance phase, the difference is always e), while in the swinging phase, we think that it should be linearly accumulated from the last zero-speed moment to the current time (from 0 up to e). Based on the above assumptions, we correct the integral data of this process.

This paper is mainly based on the proposed method, which corresponds to the obtained gait characteristics and the level results of UPDRS. How to accurately obtain the corresponding gait characteristics is not the focus of this paper, so it is not expanded in detail.

2.5.2 : Training and validation: What was the sampling frequency when you collected data from IMU? Please mention.

Thanks for your comments, we have updated the information in 2.3 Wearable System

Figure 2: Axis labelling or figure caption could be used to specify about axis, this is a minor comment.

Thanks for your comments

In discussion: It is mentioned that the results are significantly better than traditional methods: this is a huge ask,,, as if you would like to mention significant results, you need to report, also what are the traditional methods? You need to have a comparison.

We have added the results of the other three traditional models in the Discussion of latest manuscript to highlight the superiority of the proposed model

Is there any break for participants to minimise the fatigue? If that's the case, please report.

Since this experiment only requires the patient to walk no more than 20 meters in a straight line with a total of three times, according to the actual experimental situation, it will not cause the patient to experience fatigue which will obviously make the gait abnormal

Did you get any feedback from participants, especially from patients? I would suggest having a scale of 1 to 5 to mark how comfortable they were during the experiments, analyse your data and present them.

We did not carry out this part of the work, because the weight and size of the sensors are very light (we updated the relevant data in 2.3 Wearable System), and the sensor is fixed on the outer part of the human shank near the ankle, which has very few muscles. Therefore, the system does not make the human body feel uncomfortable when walking.

Conclusion: traditional methods are mentioned, what are they?

We have updated the conclusion and identify three traditional methods as Naive Bayes, support vector machine, and linear regression model

Round 2

Reviewer 3 Report

The authors have addressed most of my comments, and I, therefore, recommend acceptance of this paper.

Reviewer 4 Report

The authors addressed my comments. I am happy the progress that they made from previous version.